# Revocable and Traceable Undeniable Attribute-Based Encryption in Cloud-Enabled E-Health Systems

Zhongxiang He [1] , Yuling Chen [1,*] , Yun Luo [1] , Lingyun Zhang [2] and Yingying Tang [2]

1 State Key Laboratory of Public Big Data, Guizhou University, Guiyang 550000, China; gs.zxhe22@gzu.edu.cn (Z.H.); gs.yunluo20@gzu.edu.cn (Y.L.)
2 College of Computer Science and Technology, Guizhou University, Guiyang 550000, China; gs.lyzhang20@gzu.edu.cn (L.Z.); gs.yytang21@gzu.edu.cn (Y.T.)
* Correspondence: ylchen3@gzu.edu.cn

**Abstract:** The emerging cloud storage technology has significantly improved efficiency and productivity in the traditional electronic healthcare field. However, it has also brought about many security concerns. Ciphertext policy attribute-based encryption (CP-ABE) holds immense potential in achieving fine-grained access control, providing robust security for electronic healthcare data in the cloud. However, current CP-ABE schemes still face issues such as inflexible attribute revocation, relatively lower computational capabilities, and key management. To address these issues, this paper introduces a revocable and traceable undeniable ciphertext policy attribute-based encryption scheme (MA-RUABE). MA-RUABE not only enables fast and accurate data traceability, effectively preventing malicious user key leakage, but also includes a direct revocation feature, significantly enhancing computational efficiency. Furthermore, the introduction of a multi-permission mechanism resolves the issue of centralization of power caused by single-attribute permissions. Furthermore, a security analysis demonstrates that our system ensures resilience against chosen plaintext attacks. Experimental results demonstrate that MA-RUABE incurs lower computational overhead, effectively enhancing system performance and ensuring data-sharing security in cloud-based electronic healthcare systems.

**Keywords:** cloud; electronic healthcare; attribute-based encryption; traceable; direct revocation; multi-authority





## 1. Introduction

With the mainstreaming of cloud computing technology, cloud data sharing has become a highly regarded research topic [1,2]. Presently, the exchange of medical data is a vital endeavor aimed at improving the performance of healthcare service providers and the transformation of the healthcare system [3]. To track patients' health conditions more precisely, electronic health records (EHRs) emerged. While EHR management systems autonomously upheld by healthcare institutions do have specific constraints, this has resulted in insufficient interoperability among stakeholders [4]. Furthermore, the management mode of EHR appears to lack transparency and is also prone to internal security issues such as leaks [5]. For the assurance of confidentiality, data protection, and seamless integration of EHR data, patients can choose to employ searchable encryption methods or utilize techniques like homomorphic encryption to secure their data prior to transferring it to the cloud by employing encryption [6,7]. While this approach ensures the security of EHR data, it may struggle to meet the flexibility requirements necessary for EHR data sharing [8]. Attribute-based encryption (ABE) addresses the issue of unauthorized data access and can fulfill the need for fine-grained access control. ABE can be categorized into two forms: ciphertext policy attribute-based encryption (CP-ABE) and key policy attribute-based encryption (KP-ABE) [9,10]. KP-ABE nests the decryption key of a data user with an access policy while embedding a set of attributes into the ciphertext. In contrast, the decryption key in CP-ABE corresponds to a set of attributes, while the ciphertext of the

cloud server is associated with the access policy. Consider an EHR sharing scenario where a patient's electronic medical record is stored in the healthcare system's cloud in ciphertext with an access policy of {{Chief Physician OR Department Head} AND {Internal Medicine AND Male}}. This means that only physicians who also treat internal medicine, are male in gender, and hold the title of chief physician or department head are eligible to view patient information. This fine-grained access control ensures that only specific physicians can access sensitive medical data, thus maintaining patient privacy and data security. In contrast, CP-ABE can better address interoperability issues among stakeholders,while the owner of the EHR can flexibly adjust the embedded access policies in the ciphertext based on specific access scenarios [11]. In comparison, CP-ABE can more effectively address interoperability issues among stakeholders. However, in practical applications, CP-ABE poses risks such as key exposure and potential changes in user permissions [12,13]. Furthermore, a sole attribute authority oversees the assignment and revocation of all attributes. These schemes are vulnerable to singular points of failure, exacerbating the impact on the accessibility of attribute administration [14]. To tackle the difficulties encountered by CP-ABE, this article proposes a revocable and tracing undeniable attribute-based encryption scheme with multi-authority (MA-RUABE). Specifically, the primary contributions of the MA-RUABE scheme can be outlined as follows:

(1) Effectively tracking shared keys. A novel EHR sharing model based on cloud storage environments has been established, which can accurately identify malicious users who leak keys and build decryption devices, ensuring data protection against unauthorized access.

(2) Supports direct key revocation. By generating a special identifier binary tree for each participant and employing subset cover techniques, revocable key management has been achieved. Users who have not been revoked do not need to interact with third parties to update their keys, and this process does not affect the decryption process for other users.

(3) Adopted a strategy of power decentralization. The key generation method has been extended from single-attribute authorization to multi-attribute authorization, with collaboration among multiple authorities through secret sharing for generating global parameters, distributing keys, and managing users. This effectively prevents the misuse of private keys and mitigates the risk of single-point failures that can arise from a single authority.

(4) Ensured data non-repudiation. Users cannot deny the fact of key leakage, thus ensuring data security. Simulation experiments were conducted, and the results indicate that the MA-RUABE scheme is secure under the IND-CPA security model.

*Related Work*

In 2005, Sahai et al. [15] proposed an encryption scheme based on fuzzy identities, leading to the concept of attribute-based encryption (ABE). In 2006, Goyal et al. [9] first categorized attribute-based encryption (ABE) into cipher policy attribute-based encryption (CP-ABE) and key policy attribute-based encryption (KP-ABE). CP-ABE has had a profound impact on cloud storage technology. In practical applications, when multiple users share the same set of attributes, they can use the same key for decryption. However, this can also lead to challenges in tracing illegal sellers. Therefore, identifying the user who leaked the key becomes a crucial issue in CP-ABE. In 2008, Hinek et al. [16] first introduced the concept of traceability, which binds a user's personal information to their private key, preventing the user from leaking the key while also making it impossible to identify the specific malicious user. In 2015, Ning et al. [17] devised a white-box traceability scheme with selective plaintext security, utilizing probabilistic encryption techniques and the Shamir threshold-sharing approach to achieve traceability. Subsequently, Ning et al. [18] proposed a white-box traceable CP-ABE scheme that is fully secure under small attribute sets. This scheme employs commitment mechanisms to trace users, avoiding the need for additional identity tables. However, it may have relatively lower flexibility. In 2022, Liu et al. [19]

introduced a CP-ABE scheme with black-box accountable authority characteristics. This scheme ensures secure access and control of sensitive health data while protecting the privacy of the data. In 2023, Qu et al. [20] introduced an attribute-based traceable encryption scheme that involves equality testing and is applied in electronic health systems. However, without an effective revocation mechanism as a supplement, the utility of the traceability feature will be greatly diminished.

Regarding the revocation of user keys, the revocation mechanism can be classified into two types: direct revocation and indirect revocation, depending on the entity performing the revocation operation. In 2009, Attrapadung et al. [21] proposed a CP-ABE scheme with direct revocation, where the ciphertext is associated with the identity set of unrevealed users, leading to lower efficiency. In contrast, indirect revocation can achieve finer-grained attribute revocation and offers greater flexibility. In 2011, Hur et al. [22] introduced an indirect revocation CP-ABE scheme. Although this scheme supports attribute revocation, it is unable to effectively defend against collaborative attacks initiated by users. In 2017, Li et al. [23] proposed a novel CP-ABE scheme that requires users to possess both the system private key and attribute set key when accessing data. If a user's attributes are revoked, the system recalculates the ciphertext and attribute set key, rendering users with revoked attributes unable to decrypt the data. In 2022, Han et al. [24] combined the functionality of user revocation and hiding policies with ABE. Once a user is tracked and identified as a malicious user, its privileges will be revoked immediately. Subsequently, Ge et al. [25] presented a revocable attribute encryption with data integrity protection. This scheme is efficient and practical.

In terms of multiple-attribute authorities, in 2009, Chase and Chow [26] achieved privacy protection by preventing the certificate authority (CA) from collecting specific user information. In 2015, Li et al. [27] introduced a CP-ABE scheme with multiple-attribute authorizing authorities designed for cloud storage. However, this scheme did not incorporate user revocation functionality. In 2018, Zhu et al. [28] proposed a decentralized multi-authority CP-ABE access control scheme. This scheme achieved user revocation by distributing keys to legitimate users, but it did not overcome the issue of single-point bottleneck. In 2022, Sarma et al. [29] introduced the multi-authority scheme, where each attribute authority manages a set of mutually disjoint attributes. This scheme assigns corresponding attributes to users after verifying their roles, but it also results in increased complexity and management costs. During the same period, Zhang et al. [30] implemented a safeguard mechanism by introducing a group manager responsible for assigning certificates to individual users. This measure aimed to counteract collusion attacks involving revoked users and malicious entities. In 2023, Yan et al. [31] introduced a CP-ABE scheme with key revocation and computational outsourcing capabilities involving multiple authorities. Subsequently, Xiong et al. [32] introduced an attribute-based data-sharing scheme, granting the cloud server the capability to perform ciphertext searches. However, the scheme exhibits a lack of flexibility in attribute revocation.

The solutions mentioned earlier exhibit limitations in effectively handling key tracing, key revocation, non-repudiation, and multi-authority scenarios comprehensively. Conversely, the MA-RUABE scheme presented in this article proves to be capable of satisfying diverse security and permission requirements.

## 2. Preliminaries

### 2.1. Linear Secret-Sharing Schemes

A set of participants $\mathcal{P}$ with respect to the secret-sharing scheme $\Pi$ [33] is linear on $\mathbb{Z}_p$, and needs to satisfy the following two conditions:

1. Each participant's shared secret constitutes a column vector in $\mathbb{Z}_p$.
2. A shared generator matrix $M$ with $m$ rows and $n$ columns is associated with $\Pi$, the $i$'th row of $M$ is denoted by $\rho(i)$ and belongs to participant $i$. Considering a vector $\mathbf{v} = (s, r_2, \ldots, r_n)$, where $s$ represents the shared secret. $M_{m \times n} \cdot \mathbf{v}$ associates the $m$ shares of $\Pi$ with the secret number $s$, $\lambda_i = M_i \cdot \mathbf{v}$ is the share held by the participant $i$.

Let $\lambda_i$ be the share held by participant $i$, $\rho(i)$ be the rows in the shared generator matrix of the attributes owned by $i$. Should $i$ meet the access policy criteria, there is a constant vector $\mathbf{w}$ such that $\rho(i)^T \cdot \mathbf{w} = (1, 0, \ldots, 0)^T$, and $w_i \cdot \lambda_i = s$.

If access structure $\mathbb{A}$ has a monotonic nature, the following results follow:

- There is a vector $\mathbf{v}_1$ such that $M^T \cdot \mathbf{v}_1 = (1, 0, \ldots, 0)^T$ if $M \in \mathbb{A}$.
- There is a vector $\mathbf{v}_2$ such that $M \cdot \mathbf{v}_2 = \mathbf{0}$ if $M \notin \mathbb{A}$.

### 2.2. Composite-Order Bilinear Groups

Composite-order group bilinear mapping and prime-order group bilinear mapping have significant differences [34]. Consider three $N$-order cyclic groups $G_1, G_2, G_T$, where $N$ is the product of large prime numbers ($N = p_1 p_2 \cdots p_n$), and $p_i$ are distinct large prime numbers. For the bilinear mapping $e : G_1 \times G_2 \to G_T$, this mapping satisfies three crucial properties: linearity, non-degeneracy, and computability. Additionally, assume $G_{p_1}, G_{p_2}$, and $G_{p_3}$ are subgroups of group $G$ with orders $p_1, p_2$, and $p_3$, respectively. Choose parameters $q_i \in G_{p_i}$ and $q_j \in G_{p_j}$, where $i \neq j$, $e(q_i, q_j) = 1$.

### 2.3. Subgroup Decision Problem for Three Primes

**Assumption 1** ([35]). *Let $\mathbb{G}$ denote the order of the group, and $\mathcal{G}$ represent the group generator. Given the distribution below:*

$$\mathcal{G} \to \mathbb{G} = (N = p_1 p_2 p_3, G, G_T, e)$$

$$g_1 \leftarrow G_{p_1}, E_3 \leftarrow G_{p_3}$$

$$Distr = (E_3, g, \mathbb{G})$$

$$X_1 \leftarrow G_{p_1 p_2}, X_2 \leftarrow G_{p_1}$$

*By violating Assumption 1, algorithm $\mathcal{A}$ exhibits the following advantage:*

$$Adver1_{\mathcal{G}, \mathcal{A}}(1^\lambda) = \mid Pr[\mathcal{A}(Distr, X_1) = 1] - Pr[\mathcal{A}(Distr, X_2) = 1] \mid$$

*If $Adver1_{\mathcal{G}, \mathcal{A}}(1^\lambda)$ is a negligible function with respect to $1^\lambda$ for any polynomial-time algorithm $\mathcal{A}$, we assert that Assumption 1 is fulfilled by $\mathcal{G}$.*

**Assumption 2** ([35]). *Given the distribution below:*

$$\mathcal{G} \to \mathbb{G} = (N = p_1 p_2 p_3, G, G_T, e)$$

$$g_1, E_1 \leftarrow G_{p_1}, E_2, F_2 \leftarrow G_{p_2}, E_3, F_3 \leftarrow G_{p_3}$$

$$Distr = (\mathbb{G}, g, E_1 E_2, F_3, E_2 F_3)$$

$$X_1 \leftarrow G, X_2 \leftarrow G_{p_1 p_3}$$

*By violating Assumption 2, algorithm $\mathcal{A}$ exhibits the following advantage:*

$$Adver2_{\mathcal{G}, \mathcal{A}}(1^\lambda) = \mid Pr[\mathcal{A}(Distr, X_1) = 1] - Pr[\mathcal{A}(Distr, X_2) = 1] \mid$$

*If $Adver2_{\mathcal{G}, \mathcal{A}}(1^\lambda)$ is a negligible function with respect to $1^\lambda$ for any polynomial-time algorithm $\mathcal{A}$, we assert that Assumption 2 is fulfilled by $\mathcal{G}$.*

**Assumption 3** ([35]). *Given the distribution below:*

$$\mathcal{G} \to \mathbb{G} = (N = p_1 p_2 p_3, G, G_T, e)$$

$$\gamma, t \leftarrow \mathbb{Z}_N$$

$$g_1 \leftarrow G_{p_1}, E_2, F_2, H_2 \leftarrow G_{p_2}, E_3, F_3 \leftarrow G_{p_3}$$

$$Distr = (\mathbb{G}, g, g^{\gamma} E_2, E_3, g^t F_2, H_2)$$

$$X_1 \leftarrow e(g,g)^{\gamma t}, X_2 \leftarrow G_{p_1 p_3}$$

*By violating Assumption 3, algorithm $\mathcal{A}$ exhibits the following advantage:*

$$Adver3_{\mathcal{G},\mathcal{A}}(1^{\lambda}) = \mid Pr[\mathcal{A}(Distr, X_1) = 1] - Pr[\mathcal{A}(Distr, X_2) = 1] \mid$$

*If $Adver3_{\mathcal{G},\mathcal{A}}(1^{\lambda})$ is a negligible function with respect to $1^{\lambda}$ for any polynomial-time algorithm $\mathcal{A}$, we assert that Assumption 3 is fulfilled by $\mathcal{G}$.*

*2.4. Subset Cover*

Consider $T$ as a complete binary tree with a depth of $d$, where the leaf nodes of $T$ represent system users [36]. Let function $path(x) = (path_{x,0}, path_{x,1}, ..., path_{x,depth(x)})$ outputs the route from the root $p_{x,0} = root$ to arbitrary node $p_{x,depth(x)} = x$, and function $depth(x)$ produces the depth of node $x$. The following is the way to revoke users using the subset cover method: Marking each node in $path(x)_{\forall x \in R}$ with the revoked users set (leaf nodes) $R$. Defined as the set of unmarked nodes with direct children of marked nodes, $cover(R)$ characterizes the term. Figure 1 shows a subset cover tree, $T$ contains eight leaves $x_8, x_9, \ldots, x_{15}$. Suppose $R = \{x_{12}, x_{15}\}$, $path(x_{12}) = \{x_1, x_3, x_6, x_{12}\}$. The $cover(R)$ is defined as $\{x_2, x_{13}, x_{14}\}$. The nodes in $cover(R)$ cover the part of the node that has not been revoked from the user path.

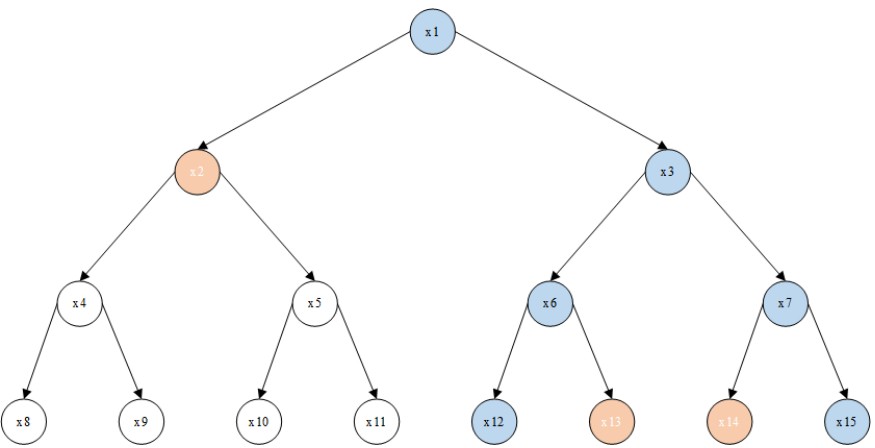

**Figure 1.** Subset cover.

## 3. MA-RUABE

*3.1. System Model*

The MA-RUABE scheme's system model comprises six entities, as depicted in Figure 2. The roles and functions of each section are outlined as follows.

- **Third-party authoritative (*TA*)**: Responsible for tracking and revoking malicious users in the system. *TA* is secure and trustworthy, capable only of generating attribute keys related to user identity. It does not have the authority to grant specific attribute meanings and cannot forge attribute keys corresponding to decentralized attribute authorities.
- **Attribute Authority (*AA*)**: Responsible for issuing meaningful attributes and generating corresponding attribute keys for EHRs. *AA* is considered semi-trusted; no individual *AA* can forge attribute keys corresponding to attributes managed by other authorization centers.
- **Cloud Service Provider (*CSP*)**: A cloud server provider is honest and inquisitive, offering data storage services.

- **Data Owner (*DO*)**: Responsible for establishing access policies to define the scope of data sharing. Patients generate ciphertext based on this access policy and transfer it to the cloud.
- **Data User (*DU*)**: Doctors receive ciphertext sent by the encryptor. They can only decrypt and obtain plaintext if the attribute key satisfies the requirements of the access policy.
- **Public Auditor (*PA*)**: In a situation where a user is suspected of key leakage, despite their claims of innocence, an audit of the user is necessary to ensure the accuracy and compliance of the entire process.

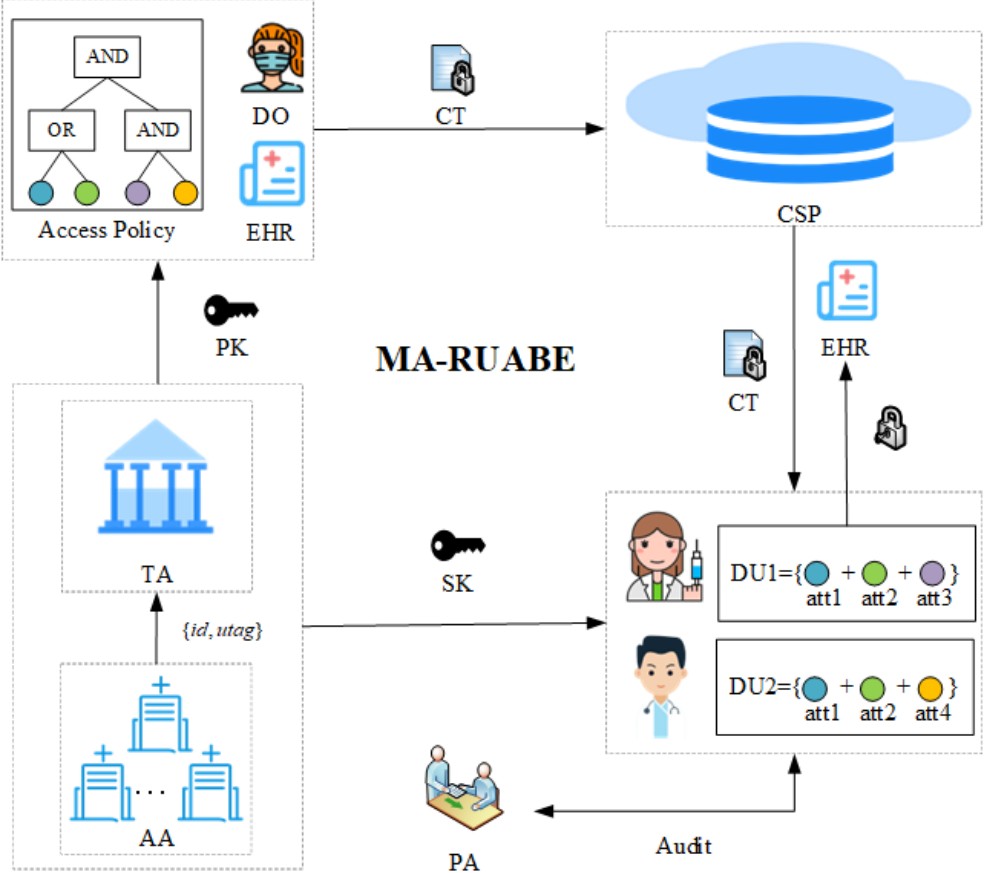

**Figure 2.** System model.

*3.2. Scheme Description*

The MA-RUABE scheme is composed of eight algorithms that run in polynomial time:

- $Setup(1^\lambda, \mathcal{U}, \mathcal{U}_I) \rightarrow (pk, msk, sk_k)$: The setup algorithm takes the secure parameters $1^\lambda$, the collective set of attributes $\mathcal{U}$ of all users in the system, and the set of user tag universe $\mathcal{U}_I$ as inputs. It generates public parameters $pk$, a master key $msk$, and private keys $sk_k$ corresponding to each attribute authority $AA_k$.
- $KeyGen(pk, msk, sk_k, id, S, utag) \rightarrow sk_{id,S,utag}$: The key generation algorithm is jointly generated by user $DU$, the authority $TA$, and each attribute authority $AA_k$ through an interactive protocol. This algorithm takes public parameter $pk$, private key $sk_k$ corresponding to each attribute authority, master key $msk$, attribute set $S \in \mathcal{U}$, user's identity $id$, and user's identifier $utag \in \mathcal{U}_I$ as inputs to generate a decryption key $sk_{id,S,utag}$.
- $Encrypt(pk, M, (A, \rho), R) \rightarrow CT_{A,R}$: The encryption algorithm requires four input parameters: public parameters $pk$, the plaintext $M$ that the user wants to encrypt, a matrix $A$ and a revocation list $R$.

- $Decrypt(pk, sk_{id,S,utag}, CT_{A,R}) \rightarrow M \ or \ ||$: The decryption algorithm takes public parameters $pk$ and the user's own decryption key $sk_{id,S,utag}$, and the ciphertext $CT_{A,R}$ is uploaded to the cloud as inputs. If the attributes of the user's key match the matrix corresponding to the access structure $A$ in the ciphertext and satisfy certain conditions $path(utag) \cap cover(R) \neq null$, then the decryption algorithm outputs the plaintext $M$.
- $KeyIntegrityCheck(pk, sk) \rightarrow 1 \ or \ 0$: The algorithm is primarily used to check whether a decryption key is complete. Public parameters $pk$ and the secret key $sk$ are used as inputs to the KeyIntegrityCheck algorithm. If $sk$ is valid, the algorithm outputs 1, otherwise, it outputs 0.
- $Trace(pk, msk, sk_k, sk) \rightarrow id$: The key tracing algorithm is primarily used to extract the user from a key and determine its ownership. Public parameters $pk$, master key $msk$, $AA'_k s$ secret key $sk_k$, and leaked key $sk$ are used as inputs to the key tracing algorithm. If the key passes the $KeyIntegrityCheck$ algorithm, the Paillier decryption algorithm is then used to extract the user's ID.
- $Audit(pk, sk_{id,S,utag}, sk^*_{id,S,utag}) \rightarrow guilty \ or \ innocent$: The Audit algorithm consists of a user and a public auditor($PA$) and is mainly used to determine the *guilty* or *innocent* of the user.
- $Update(CT_{A,R}, R') \rightarrow CT_{A,R'}$: The data owner uses an update algorithm to refresh the ciphertext, taking the original ciphertext $CT_{A,R}$ and a new revocation list $R' \supset R$ as inputs, and producing the updated ciphertext $CT_{A,R'}$ as output.

### 3.3. Security

The security of the MA-RUABE scheme is affirmed when it meets the following three criteria:

(i) The initial ciphertext's indistinguishability under chosen plaintext attack (IND-CPA).
(ii) The modified ciphertext's indistinguishability under the chosen plaintext attack.
(iii) Multiple attribute authorizations can only recover the decryption key with an ignored advantage of $\varepsilon$.

(1) The security of the initial ciphertext has been provided in reference. The definition of a security under chosen plaintext attack for the updated ciphertext is as follows:

*Setup*: The adversary $\mathcal{A}$ sends an access structure $\mathbb{A}$, a revocation lists $R$ and $R'(R \subset R')$ to challenger $\mathcal{B}$, and $\mathcal{B}$ starts the $Setup(1^\lambda, \mathcal{U}, \mathcal{U}_I)$ algorithm and sends the public parameter $pk$ to the adversary.

*Phase*1: In this phase, the adversary $\mathcal{A}$ can adaptively ask the challenger about the secret key $sk_{id_i, S_i, utag_i}$ corresponding to the user $(id_1, S_1, utag_1), (id_2, S_2, utag_2), \ldots, (id_i, S_i, utag_i), i \in [1, p_i]$. If $utag_i \notin R'$ and $S_i$ meets the access policy, the challenge is terminated, otherwise, the challenger $\mathcal{B}$ generates the decryption key $sk_{id_i, S_i, utag_i}$ through the decryption key generation algorithm $KeyGen(pk, msk, sk_k, id_i, S_i, utag_i)$, and sends $sk_{id_i, S_i, utag_i}$ to the adversary.

*Challenge*: $\mathcal{A}$ picks two messages of the same length $M_0, M_1$, an access structure $\mathbb{A}^*$ corresponds to the revocation lists $R$ and $R'$ where $R \subset R'$ and a $utag$. Note that $\mathbb{A}^*$ cannot be satisfied by any of the queried attribute sets $(id_1, S_1, utag_1), (id_2, S_2, utag_2), \ldots, (id_i, S_i, utag_i)$. The challenger flips a coin $\sigma = \{0, 1\}$ at random, runs $Encrypt(pk, M_\sigma, (A^*, \rho), R) \rightarrow CT_{A^*, R}$ and $Update(CT_{A^*, R}, R') \rightarrow CT_{A^*, R'}$, and forwards $CT_{A^*, R'}$ to $\mathcal{A}$.

*Phase*2 : $\mathcal{A}$ queries the secret key $sk_{id_i, S_i, utag_i}$ the same as in *phase*1, $i \in [p_{i+1}, p_n], S_i \notin \mathbb{A}^*$ or $utag_i \in R$.

*Guess*: $\mathcal{A}$ outputs a guess $\sigma'$, it wins this game if $\sigma = \sigma'$.

**Definition 1.** *The MA-RUABE is considered secure under a chosen plaintext attack of the updated ciphertext if a polynomial adversary can succeed in this scenario only with a negligible probability* $Pr[\sigma' = \sigma] - 1/2$.

(2) The definition of the dishonest AA game is as follows:

The game involves the interaction between the dishonest authority adversary $\mathcal{A}$ and the challenger $\mathcal{B}$. The task of adversary $\mathcal{A}$ is to attempt to recover the decryption key $sk^*_{id,S,utag}$ through this interaction to satisfy $KeyIntegrityCheck(pk, sk^*_{id,S,utag}) \rightarrow 1$ and $Trace(pk, msk, sk_k, sk) \rightarrow id$.

*Setup*: The challenger $\mathcal{B}$ generates the public parameter $pk$, the master secret key $msk$, and secret keys $sk_k$ through the $Setup(1^\lambda, \mathcal{U}, \mathcal{U}_I)$, and sends $pk$ along with the private key $sk_x$ corresponding to adversary $\mathcal{A}$ to $\mathcal{A}$.

*Phase*: $\mathcal{A}$ queries $\mathcal{B}$ for the decryption key of any user $(id, S, utag)$. $\mathcal{B}$ first generates a portion $sk_{pri}$ of the decryption key, computes $sk_{id,S,utag}$ using the $Decrypt$ algorithm, then sends the generated parameters to $\mathcal{A}$, and retains $sk_{id,S,utag}$.

*Challenge*: $\mathcal{A}$ attempts to recover a decryption key $sk^*_{id,S,utag}$ based on the parameters sent by challenger $\mathcal{B}$.

**Definition 2.** *We call a scheme multi-attribute and authoritatively secure if, for any polynomial-time dishonest adversary $\mathcal{A}$, the game can be won only with negligible probability $Pr[KeyIntegrityCheck(pk, sk^*_{id,S,utag}) \rightarrow 1$ and $Trace(pk, msk, sk_k, sk) \rightarrow id] < \varepsilon$.*

## 4. Specific Construction of MA-RUABE

### 4.1. Construction

- $Setup(1^\lambda, \mathcal{U}, \mathcal{U}_I) \rightarrow (pk, msk, sk_k)$: The setup algorithm produces an order $N = p_1 p_2 p_3$ bilinear group $G$ through the group generator $\mathcal{G}$, and $p_1, p_2, p_3$ are three distinct primes. $G_{p_i}$ is of order $p_i$ in $G's$ subgroup. $g, g_3$ are generators of $G_{p_1}, G_{p_3}$ respectively, defining a mapping $e: G \times G \rightarrow G_T$, then the algorithm chooses random elements $\alpha, m, a, b, c, d \in \mathbb{Z}_N$, and it selects random values $u_i, \beta_i \in \mathbb{Z}_N$ for each attribute $i \in \mathcal{U}$. Also, the algorithm randomly selects $p, q(p \neq q, p \text{ and } q \text{ have the same length})$, and $gcd(pq, (p-1)(q-1)) = 1$, let $\pi = lcm(p-1, q-1), n = pq, Q = \pi^{-1} \mod n$, $g_1 = (1+n)$. Moreover, it takes a hash function $F: \mathcal{U}_I \rightarrow \mathbb{Z}_N$, sets

$$path(utag) = (p_{utag,0}, p_{utag,1}, \ldots, p_{utag,d})$$

$d$ represents the height of the full binary tree, where $p_{utag,0} = root$ and $p_{utag,d} = utag$. The public parameters

$$pk = (N, n, g_1, g, g^a, g^b, g^c, g^d, g^m, e(g,g)^\alpha, \{\forall utag \in \mathcal{U}_I, g^{F(x_r)}\}_{x_r \in path(utag)},$$

$$\{\mathcal{U}_i = g^{u_i}, \mathcal{V}_i = g^{\beta_i}\}_{i \in \mathcal{U}})$$

$msk = (p, q, \alpha, a, g_3)$ and secret key $sk_k = \{\beta_i\}_{i \in AA_k}$ corresponding to the authorized agency $AA_k$.

- $KeyGen(pk, msk, sk_k, id, S, utag) \rightarrow sk_{id,S,utag}$: The key generation algorithm is jointly generated by the user $DU$, the authority $TA$, and each attribute authority $AA_k$ through an interactive protocol:

1. $DU$ sends its own attributes $\{s_i\}_{i \in AA_k}$ to organization $AA_k$, which has the authorization authority for the corresponding attributes.

2. $AA_k$ calculates $\bar{D}_i = \{\mathcal{U}_i^{\beta_i}\}_{i \in AA_k}$ and sends $\bar{D}_i$ to $DU$.

3. $DU$ first verifies the following equation:

$$\text{for}\{s_i\}_{i \in S}, \text{there is } e(\mathcal{V}_i, \mathcal{U}_i) = e(g, \bar{D}_i)$$

If the equation holds, $DU$ randomly selects $x, y \in \mathbb{Z}_N$ and calculates $t = xy$, $R_U = g^t$, then sends $g^t$, identity $id$, unique identifier $utag$ and $\{\bar{D}_i\}_{i \in S}$ to $TA$, then runs an interactive zero-knowledge proof of $R_U$ about $t$.

4. *TA* first verifies whether $R_U$ is generated by $t$, if the verification passes, *TA* randomly selects $h \in \mathbb{Z}_N, k \in \mathbb{Z}_n^*$ and random elements $R_0, R_1, R_2, R_3, \{R_{x_r}\}_{x_r \in path(utag)},$ $\{R_i', R_i''\}_{i \in S} \in G_{p_3}$, then *TA* calculates a part of the decryption key:

$$sk_{pri} = <\bar{D}_0 = g^{\frac{\alpha}{a+\bar{T}}}(R_U)^{\frac{b}{a+\bar{T}}}g^{dh}R_0, \bar{T} = g_1^{id}k^n mod n^2,$$

$$\bar{D}_1 = g^h R_1, \bar{D}_2 = g^{mh}R_2, \bar{D}_3 = g^{ah}R_3,$$

$$\{\bar{D}_{x_r} = g^{hF(x_r)}R_{x_r}\}_{x_r \in path(utag)},$$

$$\{\bar{G}_{i,1} = \mathcal{U}_i^{\beta_i h(a+\bar{T})}R_i', \bar{G}_{i,2} = \mathcal{V}_i^{(a+\bar{T})h}R_i''\}_{i \in S} >$$

It then sends $(h, sk_{pri})$ to *DU*.

5. *DU* initially checks if the following equation is valid:

   (1) $e(\bar{D}_1, g^a) = e(\bar{D}_3, g) = e(g, g)^{ah}.$

   (2) $e(\bar{D}_0, g^a g^T) = e(g, g)^\alpha e(R_U, g^b)e((\bar{D}_1)^T \bar{D}_3, g^d).$

   (3) $\exists x \in S, s.t. e(\mathcal{U}_x, \bar{G}_{x,2}) = e(\bar{G}_{x,1}, g), e(\bar{D}_{x,1}, g^a) = e(g_x^\beta, \bar{D}_3).$

   If the equation holds, *DU* calculates $t_{id} = \frac{h}{t}$ and generates the decryption key:

   $$sk_{id,S,utag} = <S, D_0 = \bar{D}_0(g^c)^{t_{id}}, T = \bar{T}, D_1 = \bar{D}_1,$$

   $$D_2 = \bar{D}_2, D_3 = \bar{D}_3, \{D_{x_r} = \bar{D}_{x_r}\}_{x_r \in path(utag)}, t_{id}, R_U,$$

   $$\{G_{i,1} = \bar{G}_{i,1}, G_{i,2} = \bar{G}_{i,2}\}_{i \in S} >$$

   We distribute the attributes to different institutions. *TA* lacks access to the secret key $\beta_i$, and $AA_k$ is not aware of *TA*'s *msk*. Therefore, only a few institutions are unable to recover the decryption key.

- *Enctypt$(pk, M, (A, \rho), R) \to CT_{A,R}$*: The encryption algorithm first encodes the access structure $A$ with *LSSS* scheme, and then selects a vector $\mathbf{y} = (s, y_2, \ldots, y_n)$, where $s$ is the shared secret number and $y_2, \ldots, y_n \in \mathbb{Z}_N$ is randomly selected, then selects random elements $x_i, r_i \in \mathbb{Z}_N$ for each row of the matrix M. Define $[l] = 1, \ldots, m$, where $m$ denotes the number of rows of the matrix. The ciphertext is composed of the following:

  $$CT_{A,R} = <C = M \cdot e(g, g)^{\alpha s}, C_0 = g^s, C_1 = (g^a)^s, C_2$$

  $$= (g^b)^s, C_3 = (g^c)^s, C_4 = (g^m)^s,$$

  $$\{C_{x_r} = (g^{F(x_r)})^s\}_{x_r \in cover(R)}, \{C_{i,1} = g^{dA \cdot y}\mathcal{V}_i^{-x_i},$$

  $$C_{i,2} = g^{x_i}, C_{i,3} = g^{r_i}, C_{i,4} = \mathcal{U}_i^{-r_i}\}_{i \in [l]}, (A, \rho) >$$

- *Decrypt$(pk, sk_{id,S,utag}, CT_{A,R}) \to M \ or \ ||$*: The algorithm takes the user's decryption key $sk_{id,S,utag}$, ciphertext $CT_{A,R}$, and public parameter *pk* as input, if $S$ satisfies the access structure and $utag \notin R$. It first calculates the vector $\mathbf{w} = (w_i)$ so that $\sum_{\rho(i) \in S} w_i A_i^T = (1, 0, \ldots, 0)$, and if user $i \notin R$, then there is an $x_r = cover(R) \cap path(utag)$ such that $F(x_r)_{x_r \in path(utag)} = F(x_r)_{x_r \in cover(R)}$, then calculates:

  $$D = (e((C_0)^T C_1, D_0)e(D_{x_r}, C_4))(e(C_2, R_U)e(C_3, (g^T g^a)^{t_{id}})e(D_2, C_{x_r}))^{-1}$$

  $$E = \Pi_{\rho(i) \in S}(e(C_{i,1}, D_1^T D_3)e(C_{i,3}, G_{i,1})e(C_{i,2}C_{i,4}, G_{i,2}))^{w_i}$$

  plaintext $M = \frac{C}{D/E}$.

- *KeyIntegrityCheck$(pk, sk) \to 1 \ or \ 0$*: The algorithm takes public parameter *pk* and a decryption key *sk* as input, and the *sk* is valid if:

1. *sk* is expressed as

$$(S, D_0, T, D_1, D_2, D_3, \{D_{x_r}\}_{x_r \in path(utag)}, R_U, t_{id}, \{G_{i,1}, G_{i,2}\}_{i \in S})$$

and $S, D_0, D_1, D_2, D_3, \{D_{x_r}\}_{x_r \in path(utag)}, R_U, t_{id}, \{G_{i,1}, G_{i,2}\}_{i \in S} \in G, T \in \mathbb{Z}_{n^2}^*$.

2. $e(D_1, g^a) = e(D_3, g) = e(g, g)^{ah}$.
3. $e(D_0, g^a g^T) = e(g, g)^\alpha e((D_1)^T D_3, g^d) e(R_U, g^b) e((g^a g^T)^{t_{id}}, g^c)$.
4. $\exists x \in S, s.t. e(\mathcal{U}_x, G_{x,2}) = e(G_{x,1}, g)$.
5. $\forall x_r \in path(utag), s.t. e(D_{x_r}, g^m) = e(g^{F(x_r)}, D_2)$.

- *Trace*$(pk, msk, sk_k, sk) \rightarrow id$: After the key successfully passes the *KeySanityCheck* algorithm, the *Trace* algorithm can decrypt the Paillier encryption and extract the *id* from the key.

- *Audit*$(pk, sk_{id,S,utag}, sk_{id,S,utag}^*) \rightarrow guilty\ or\ innocent$: When a user is suspected of being guilty, but he himself claims to be innocent, *DU* interacts with the public auditor *PA*:

  1. *DU* provides its decryption key $sk_{id,S,utag}$ to the public auditor *PA*, and if it passes the *KeyIntegrityCheck* algorithm, proceeds to the second step.
  2. *PA* verifies whether $t_{id} = t_{id}^*$. As our scheme employs multiple authoritative institutions to issue decryption keys, only a few entities are unable to recover the key. If this equation holds, then *DU* cannot deny the fact that it leaked the decryption key.

- *Update*$(CT_{A,R}, R') \rightarrow CT_{A,R'}$: The key update algorithm takes the original ciphertext $CT_{A,R}$, a revocation list $R'$ as input, and publishes $R'$ publicly, as shown in Figure 3.

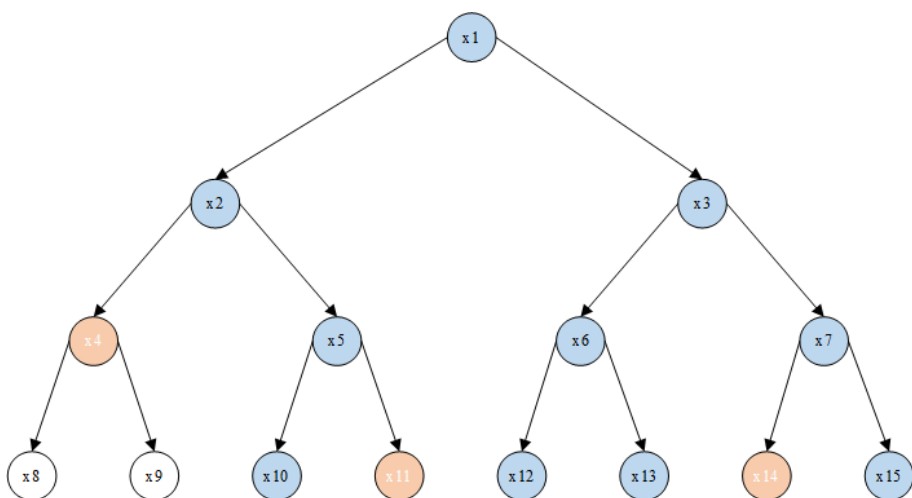

**Figure 3.** Updated subset cover.

Assuming that the revocation list is $\{x_{10}, x_{13}\}$, then $cover(R') = \{x_4, x_{11}, x_{14}\}$, and the data owner modifies the ciphertext. $CT_{A,R'}$ according to the revocation list is as follows:

$$CT_{A,R'} =< \tilde{C} = C, \tilde{C}_1 = C_1, \tilde{C}_2 = C_2, \tilde{C}_3 = C_3, \tilde{C}_4 = C_4,$$

$$\{\tilde{C}_{x_r} = g^{mF(x_r)s}\}_{x_r \in cover(R')}, \{\tilde{C}_{i,1} = C_{i,1}, \tilde{C}_{i,2} = C_{i,2},$$

$$\tilde{C}_{i,3} = C_{i,3}, \tilde{C}_{i,4} = C_{i,4}\}_{i \in [l]}, (A, \rho) >$$

### 4.2. Correctness

If a user is not included in the revocation set, then there is an $x_j = cover(R) \cap path(utag)$ such that $F(x_j)_{x_j \in path(utag)} = F(x_j)_{x_j \in cover(R)}$, and $R_0, R_2, R_{x_r} \in G_{p_3}$. In accordance with the orthogonal characteristic of composite-order bilinear groups:

$$D = \frac{e(g^{s(a+T)}, g^{ct_{id}})e(g^{s(a+T)}, g^{\frac{\alpha}{a+T}})e(g^{s(a+T)}, g^{\frac{bt}{a+T}})}{e((g^b)^s, g^t)} \cdot \frac{e((g^c)^s, (g^T g^a)^{t_{id}})}{e(g^{s(a+T)}, g^{dh})}$$

$$= e(g^s, g^{\alpha})e(g^{s(a+T)}, g^{dh})$$

If a user is included in the revocation set:

$$D = \frac{e((g^s)^T (g^a)^s, g^{ct_{id}} g^{\frac{\alpha}{a+T}} g^{\frac{bt}{a+T}} g^{dh} R_0)}{e((g^b)^s, g^t)e((g^c)^s, (g^T g^a)^{t_{id}})} \cdot \frac{e(g^{hF(x_r)} R_{x_r}, (g^m)^s)_{x_r \in path(utag)}}{e(g^{mh} R_2, (g^{F(x_j)})^s)_{x_j \in cover(R)}}$$

$$E = \Pi_{\rho(i) \in S}(e(g^{dA \cdot \mathbf{y}} \mathcal{V}_{\rho(i)}^{-x_{\rho(i)}}, (g^h R_3)^T g^{ah} R_4)e(g^{r_{\rho(i)}}, \mathcal{U}_{\rho(i)}^{\beta_{\rho(i)}(a+T)h} R_i)e(g^{x_{\rho(i)}} \mathcal{U}_{\rho(i)}^{-r_{\rho(i)}}, \mathcal{V}_{\rho(i)}^{(a+T)h}))^{w_i}$$

$$= \Pi_{\rho(i) \in S}(e(g^{dA \cdot \mathbf{y}}, g^{h(a+T)})e(\mathcal{V}_{\rho(i)}^{-x_{\rho(i)}}, g^{h(a+T)})e(g^{r_{\rho(i)}}, \mathcal{U}_{\rho(i)}^{\beta_{\rho(i)}(a+T)h})e(g^{x_{\rho(i)}}, \mathcal{V}_{\rho(i)}^{(a+T)h})$$

$$e(\mathcal{U}_{\rho(i)}^{-r_{\rho(i)}}, \mathcal{V}_{\rho(i)}^{(a+T)h}))^{w_i}$$

$$= \Pi_{\rho(i) \in S}(e(g^{dA \cdot \mathbf{y}}, g^{h(a+T)}))^{w_i}$$

$$= e(g, g)^{dh(a+T) \sum_{\rho(i) \in S} (A \cdot \mathbf{y})^T \cdot w_i}$$

$$= e(g, g)^{dh(a+T)s}$$

$$D/E = e(g, g)^{\alpha s}, M = \frac{C}{D/E}$$

### 4.3. IND-CPA Security

The literature has demonstrated the security of the initial ciphertext. After the ciphertext has been updated, then we demonstrate the IND-CPA security. First, a semi-functional ciphertext (S-FC) and semi-functional keys (S-FK) [37] must be created:

Given revocation lists $R, R'(R \subset R')$, randomly select $f \in \mathbb{Z}_N$, $g_2$ as the generator of $G_{p_2}$. Randomly choose $z_i, w_i \in \mathbb{Z}_N$ for attributes, and select elements $\gamma_i, v_i \in \mathbb{Z}_N$ along with a vector $\mathbf{u} \in \mathbb{Z}_N$. The definition of the S-FC after updating the ciphertext is as follows:

$$\tilde{C}_0 = g^s g_2^f, \tilde{C}_1 = g^{as} g_2^f, \tilde{C}_2 = g^{bs} g_2^f, \tilde{C}_3 = g^{cs} g_2^f,$$

$$\tilde{C}_4 = g^{ms} g_2^{2f}, \{\tilde{C}_{x_r} = (g^{F(x_r)})^s g_2^f\}_{x_r \in cover(R')},$$

$$\{\tilde{C}_{i,1} = g^{dA \cdot \mathbf{y}} \mathcal{V}_{\rho(i)}^{-x_i} \cdot g_2^{A \cdot \mathbf{u} + \gamma_i w_{\rho(i)}}, \tilde{C}_{i,2} = g^{x_i} g_2^{-\gamma_i},$$

$$\tilde{C}_{i,3} = g^{r_i} g_2^{-v_i}, C_{i,4} = \mathcal{U}_i^{r_i} g_2^{-v_i z_{\rho(i)}}\}$$

Randomly select $h, k$ to define the following two S-FKs:

$$Type1 : \tilde{D}_0 = D_0 \cdot g_2^h, \tilde{D}_1 = D_1 \cdot g_2^k, \tilde{D}_2 = D_2 \cdot g_2^{k+h},$$

$$\tilde{D}_3 = D_3 \cdot g_2^{kT}, \tilde{t}_{id} = t_{id}, \tilde{R}_U = R_U, \tilde{D}_{x_r} = D_{x_r} \cdot g_2^k,$$

$$\tilde{G}_{i,1} = G_{i,1} \cdot g_2^{2kTz_i w_i}, \tilde{G}_{i,2} = G_{i,2} \cdot g_2^{2kTw_i}$$

$$Type2 : \tilde{D}_0 = D_0 \cdot g_2^h, \tilde{T} = T, \tilde{D}_1 = D_1, \tilde{D}_2 = D_2,$$

$$\tilde{D}_3 = D_3, \tilde{R}_U = R_U, \tilde{t}_{id} = t_{id}, \tilde{D}_{x_r} = D_{x_r}, \tilde{G}_{i,1} = G_{i,1},$$

$$\tilde{G}_{i,2} = G_{i,2}(let \; k = 0)$$

The S-FK can only decrypt the S-FC, but the ordinary key can also decrypt the ordinary ciphertext. There will be an extra item when we use an S-FK to decrypt the S-FC:

$$e(g_2, g_2)^{T(fh - 2u_1 k)}$$

Through a sequence of games, we demonstrate the security of the MA-RUABE system:

- **Game**$_{real}$: The keys and ciphertexts used in this simulation of a security game are standard.
- **Game**$_0$: In this stage, all keys are common, and the ciphertext is only semi-functional.
- **Game**$_{k,1}$: The challenge ciphertext and first $k-1$ keys of Type2 and the $k$-th key of $Type1$ are both semi-functional.
- **Game**$_{k,2}$: The challenge ciphertext in this game is S-FC, and the first $k$ keys are S-FK of $Type2$, with the remaining keys being common keys.

In the final stage of the game, we engage in the last round of the game($Game_{final}$): all of the keys are $Type2$ semi-functional keys, and the ciphertext is produced by semi-functionally encrypting.

**Lemma 1.** *Assuming there is a polynomial algorithm $\mathcal{A}$ such that $Game_{real} Adv_{\mathcal{A}}$-$Game_0 Adv_{\mathcal{A}}=\varepsilon$, we can construct an algorithm in polynomial time to break Assumption 1 with the advantage of $\varepsilon$.*

**Proof.** Send $\alpha, a, g_3, \beta_i$ to $\mathcal{B}$, he will simulate **Game**$_{real}$ and **Game**$_0$ with $\mathcal{A}$. $\mathcal{A}$ sends an access structure $(\mathbb{A}^*, \rho)$ and revocation lists $R, R'(R \subset R')$ to $\mathcal{B}$. $\mathcal{B}$ randomly selects exponents $\alpha, m, a, b, c, d \in \mathbb{Z}_N$, and selects $u_i, \beta_i$ for each attribute $i$ in the system, a function $F : \mathcal{U}_I \to \mathbb{Z}_N$, and then sends the public parameter $pk = (N, n, g_1, g, g^a, g^b, g^c, g^d, g^m, e(g,g)^{\alpha}, \{\forall utag \in \mathcal{U}_I, g^{F(x_r)}\}_{x_{r(utag)}}, \{\mathcal{U}_i = g^{u_i}, \mathcal{V}_i = g^{\beta_i}\}_{i \in \mathcal{U}})$ to $\mathcal{A}$. $\mathcal{A}$ sends two plaintexts $M_0, M_1$ of equal length to $\mathcal{B}$, and $\mathcal{B}$ implicitly sets $g^s$ the $G_{p1}$ part of $T$. $\mathcal{B}$ chooses $\beta = \{0,1\}$ by tossing a coin, and sets the ciphertext in the following format:

$$C = M_{\beta} \cdot e(g^{\alpha}, T), \tilde{C}_0 = T, \tilde{C}_1 = T^a, \tilde{C}_2 = T^b,$$

$$\tilde{C}_3 = T^c, \tilde{C}_4 = T^m, \{\tilde{C}_{x_r} = T^{F(x_r)}\}_{x_r \in cover(R')}.$$

$\mathcal{B}$ randomly selects $\{y'_2, \ldots, y'_n\} \in \mathbb{Z}_N$, sets $y' = (1, y_2, \ldots, y_n)$, randomly selects random values $x_i, r_i$, for each row of $A^*$, and sets

$$\tilde{C_{i,1}} = T^{dA^* \cdot \mathbf{y}'} T^{-x'_i \beta_{\rho(i)}}, \tilde{C_{i,2}} = T^{x'_i}, \tilde{C_{i,3}} = T^{r'_i}, \tilde{C_{i,4}} = T^{\beta_{\rho(i)} u_i r'_i}.$$

$\mathcal{B}$ implicitly sets $\mathbf{y}$ to $(s, sy'_2, \ldots, sy'_n)$, $x_i = sx'_i, r_i = sr'_i$, because of $g^s, g^a, g^b, g^c, g^d, g^m \in G_{p1}$.

If $T \in G_{p1}$, this is a normal ciphertext after the update.

If $T \in G_{p1p2}$, let $g_2^f$ be the part of $G_{p2}$ in $T$, where $T = g^s g_2^f$. Let

$$\mathbf{u} = fd \cdot \mathbf{y}', \gamma_i = -(f \cdot x'_i)_{\rho(i) \in S}, w_{\rho(i)} = \beta_{\rho(i)}, r_i = -(f \cdot r'_i), z_{\rho(i)} = u_{\rho(i)}.$$

This is a uniformly distributed semi-functional ciphertext. Therefore, the game can be won by $\mathcal{A}$ with the advantage of $\varepsilon$. Since it is only different from the ciphertext structure in [17,37], Assumptions 2 and 3 can be obtained by the above construction and the proof. □

Dishonest Attribute Authority Game

**Lemma 2.** *We can create an algorithm $\mathcal{B}$ in polynomial time to disprove Assumption 4 with the advantage of $\varepsilon$, assuming there is a polynomial algorithm $\mathcal{A}$ such that $Adv_{\mathcal{A}} = \varepsilon$.*

**Proof.** The challenger $\mathcal{B}$ starts the Setup algorithm to generate the public parameter, the master secret key, and secret keys, where $pk = (N, n, g_1, g, g^a, g^b, g^c, g^d, g^m, e(g,g)^{\alpha}, \{\forall utag \in$

$\mathcal{U}_I, g^{F(x_r)}\}_{x_{r(utag)}}, \{\mathcal{U}_i = g^{u_i}, \mathcal{V}_i = g^{\beta_i}\}_{i \in \mathcal{U}}), msk = (p, q, \alpha, a, g_3), sk_k = \{\beta_i\}, \mathcal{B}$ sends $pk$ to adversary $\mathcal{A}$. $\mathcal{A}$ asks $\mathcal{B}$ about the decryption key of user $(id, utag, S)$. $\mathcal{B}$ generates part of the decryption key:

$$sk_{pri} = < \bar{D}_0 = g^{\frac{\alpha}{a+\bar{T}}} g^{\frac{b}{a+\bar{T}}} g^{dh} R_0, \bar{T} = g_1^{id} k^n mod n^2, \bar{D}_1 = g^h R_1, \bar{D}_2 = g^{mh} R_2, \bar{D}_3 = g^{ah} R_3,$$

$$\{\bar{D}_{x_r} = g^{hF(x_r)} R_{x_r}\}_{x_r \in path(utag)}, \{\bar{G}_{i,1} = \mathcal{U}_i^{\beta_i h(a+\bar{T})} R_i', \bar{G}_{i,2} = \mathcal{V}_i^{(a+\bar{T})h} R_i''\}_{i \in S} >$$

$\mathcal{B}$ randomly selects $x', y' \in \mathbb{Z}_N$ and sets the decryption key:

$$sk_{id,S,utag} = < S, D_0 = g^{\frac{\alpha}{a+\bar{T}}} g^{\frac{bx'y'}{a+\bar{T}}} g^{dh} R_0 g^{\frac{ch}{x'y'}}, T = \bar{T}, D_1 = \bar{D}_1, D_2 = \bar{D}_2,$$

$$D_3 = \bar{D}_3, \{D_{x_r} = \bar{D}_{x_r}\}_{x_r \in path(utag)}, t_{id}, R_U, \{G_{i,1} = \bar{G}_{i,1}, G_{i,2} = \bar{G}_{i,2}\}_{i \in S} >$$

Then $\mathcal{B}$ sends $sk_{pri}, g^{x'}, g^{y'}, h$ to $\mathcal{A}$. $\mathcal{A}$ tries to obtain the value of $g^{x'y'}$ through $g^{x'}, g^{y'}$. After calculating, $\mathcal{A}$ selects $m', n' \in \mathbb{Z}_N$, sets $t_{id}^* = h/m'n'$, $R_U^* = g^{m'n'}$, and generates the decryption key. At this time, the *KeyIntegrityCheck* algorithm outputs 1, and the *Trace* algorithm outputs $id$.

$$Pr[Audit \to 1] = Pr[t_{id} = t_{id}^*] = Pr[h/m'n' = h/x'y'] = Pr[g^{m'n'} = g^{x'y'}] = \varepsilon.$$

Since the *CDH* assumption is an *NP* problem, adversary $\mathcal{A}$ can therefore break Assumption 4 with the advantage of $\varepsilon$. $\square$

## 5. Comparsion

### 5.1. Property Comparison

As shown in Table 1, for tracking overhead, TR-APABE [24] requires maintaining an identity table and performing corresponding identity searches in this table every time the tracking algorithm is executed. The scheme RABE-DI [25] allows for the updating the access policy for ciphertexts, but does not enable direct user revocation. On the contrary, TLU-CPABE [17] and MA-RUABE only have to retain a constant value $k$ to achieve traceability. However, both schemes assume that the central authority is completely trusted and susceptible to attacks from a corrupt central authority. G-ABEET [32] is an extension of KP-ABE, but the EHR's attributes visitors typically remain stable. Therefore, EHR owners need to adjust the embedded access policies based on the access scenario. In comparison, MA-RUABE is the only solution that achieves the multi-attribute property, traceability, and attribute revocation in an adaptive secure manner, ensuring that users' data privacy in the electronic healthcare environment is protected from various threats.

**Table 1.** Comparison of MA-RUABE scheme and other schemes.

| Scheme | TLU-CPABE | TR-APABE | RABE-DI | G-ABEET | MA-RUABE |
|---|---|---|---|---|---|
| Type of ABE | CP-ABE | CP-ABE | CP-ABE | KP-ABE | CP-ABE |
| Access Structure | LSSS | LSSS | LSSS | LSSS | LSSS |
| Key Revocation | × | ✓ | ✓ | × | ✓ |
| Adaptive Security | ✓ | × | × | × | ✓ |
| Traceability | ✓ | ✓ | × | × | ✓ |
| Multiple Authority | × | × | × | ✓ | ✓ |

### 5.2. Efficiency Comparison

To perform a thorough analysis of the feasibility and effectiveness of this scheme, this section employs simulation experiments to compare the performance of various schemes. We utilize the Java-based JPBC library to construct the scheme and evaluate the efficiency of the encryption scheme. The experiments are conducted on a Windows 11 system platform with 16 GB of RAM, equipped with a six-core R5-2600 processor operating at a frequency

of 3.40 GHz. The composite-order bilinear group is configured with a size of 128 bits, and the attribute set's size increases exponentially, taking values of 2, 4, 8, and so on.

In the private key generation phase, as shown in Figure 4a, as the attributes associated with the key increase, the key size, and generation time exhibit linear expansion. TR-APABE stands out as the most efficient solution during this phase, demonstrating the shortest key generation time and minimal key size. Our proposed scheme shares the same level of efficiency as TR-APABE.

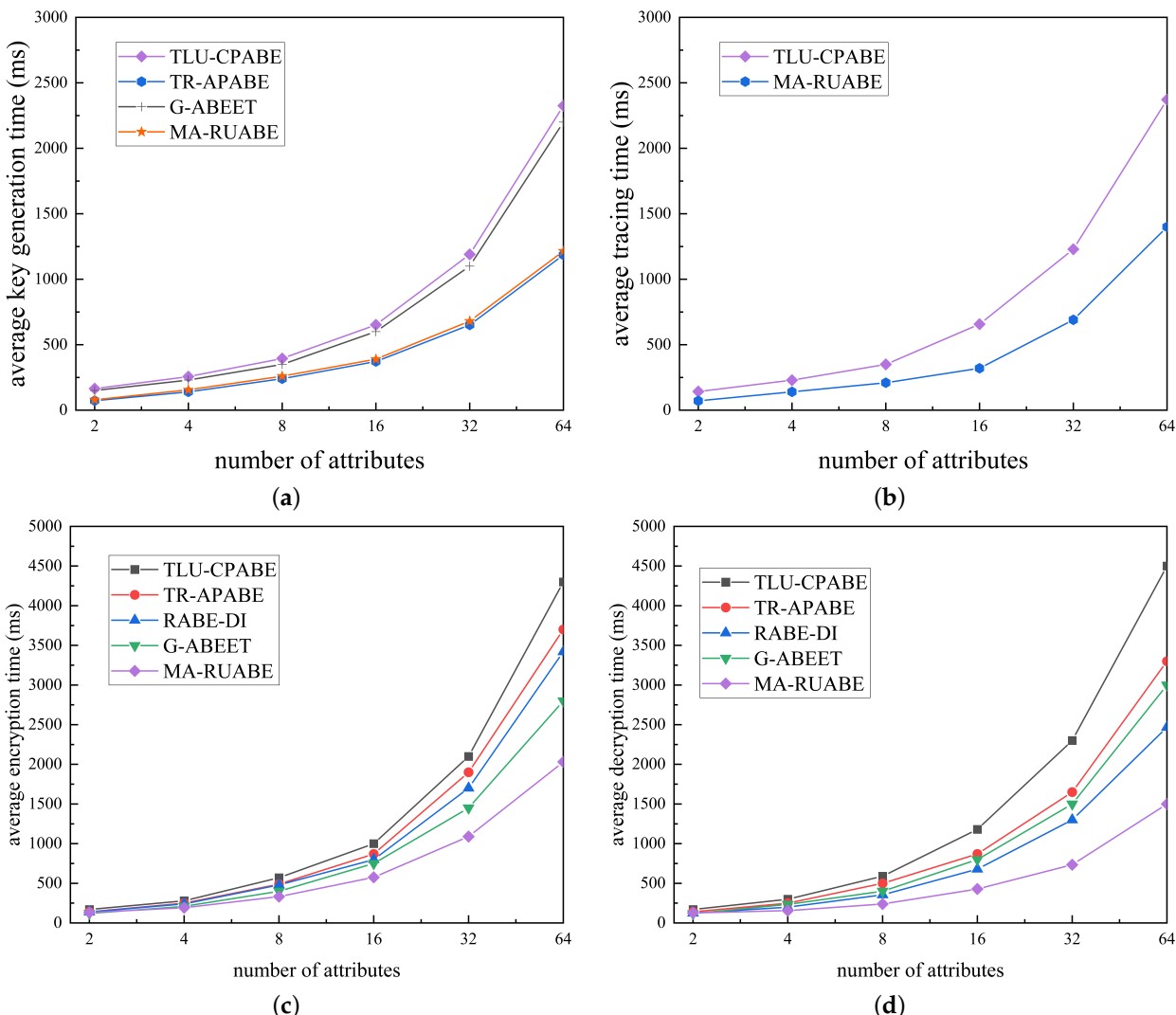

**Figure 4.** Time of encryption and decryption. (**a**) Key generating time; (**b**) tracing time; (**c**) encryption time; (**d**) decryption time.

In the tracking phase, as shown in Figure 4b, compared to TLU-ABE, MA-RUABE exhibits a certain advantage in traceability effectiveness.

In the encryption phase, as shown in Figure 4c, with an increase in the number of attributes associated with the ciphertext, both the size of the ciphertext and the encryption time exhibit linear growth. Although this scheme introduces subset coverage technology, the complexity in parameter selection remains $O(N)$. Therefore, compared to previous schemes in this stage, the suggested scheme showcases superior efficiency, characterized by the briefest encryption time. However, ciphertext construction is relatively complex, resulting in a marginally greater size of ciphertext.

In the decryption phase, as illustrated in Figure 4d, this scheme requires an intersection operation on a set, but the time required for this step can be considered negligible. Hence,

relative to previous schemes, the proposed scheme is also the most effective in this stage, boasting the shortest decryption time. Furthermore, both TR-APABE and G-ABEET incur additional search costs, which escalate with the growing number of users.

In summary, MA-RUABE represents a reliable data privacy protection scheme, exhibiting outstanding performance in cloud-based electronic healthcare environments. It demonstrates both practicality and efficiency.

## 6. Conclusions and Future Work

To accomplish efficient data sharing in the electronic healthcare cloud environment, we have introduced a revocable and traceable undeniable adaptively secure scheme (MA-RUABE), based on TLU-CPABE. This scheme employs subset coverage techniques and multi-authority key distribution to effectively address the potential misuse of keys resulting from malicious key sharing by users. It also ensures that the decryption process for other members of the system remains unaffected. Experimental evaluations demonstrate that MA-RUABE provides both high efficiency and sufficient security, effectively safeguarding data sharing within the electronic healthcare cloud system.

One future direction is to further optimize the proposed scheme and enhance the current architecture. This involves standardizing the system model and continuously improving it to bolster the overall resilience of the system. The goal is to advance the system's intelligence and adaptability. Additionally, a crucial direction involves integrating the scheme with other advanced technologies, particularly incorporating blockchain technology. By introducing blockchain, the security and functionality of the MA-RUABE scheme can be further strengthened to address emerging challenges in the electronic healthcare cloud environment.

**Author Contributions:** Z.H. designed the article structure, composed the manuscript, and performed the experimental tests. Supervisor Y.C. provided financial backing. Y.L., L.Z. and Y.T. critically reviewed the manuscript. All authors have read and agreed to the published version of the manuscript.

**Funding:** This research is funded by the National Natural Science Foundation (61962009), (62202118). Natural Science Research Technology Top Talent Project of Guizhou Provincial Department of Education (Qianjiao ji [2022]073), Science and Technology Tackling Project of Guizhou Education Department (Qianjiao ji [2023]003), and Hundred-level Innovative Talent Project of Guizhou Provincial Science and Technology Department (Qiankehe Platform Talent-GCC[2023]018).

**Institutional Review Board Statement:** Not applicable.

**Data Availability Statement:** Not applicable.

**Conflicts of Interest:** The authors declare no conflicts of interest.

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
