# Peer review of "Revocable and Traceable Undeniable Attribute-Based Encryption in Cloud-Enabled E-Health Systems"

_entropy, doi:10.3390/e26010045_

Round 1
Reviewer 1 Report
Comments and Suggestions for Authors
The comments are given as follows.
1. The main problem is that the contribution of the paper seems minor. The major part of the proposed scheme is from [17], that is also the reason the authors states the IND-CPA security based on the results of [17]. To achieve the revocability, the authors combine a well-studied skill, i.e. subset cover framework, into the scheme of [17] to obtain their scheme. Unfortunately, it seems that such combination is straightforward. The authors should give the technical breakthrough of their scheme, and show that such combination is not trivial. Besides, it is known that white-box tracing for ABE is trivial, since a key is indeed a signature generated by AA, and hence it can be verified using AA's public parameter.
2. The definition and the security model are not clear. The authors attempt to provide a multi-authority framework in the paper. However, in the definition shown in Section 3.2, there is only a Setup algorithm, which means the entity perform this algorithm will know the master secret of the system. This implies the definition is a centralized framework. Besides, if the authors claim that their scheme is a multi-authority ABE scheme, then they should consider the collusion between malicious AAs in the security model shown in Section 3.3.
3. The proposed scheme does not meet the definition shown in Section 3.2. For example, there is no KeyIntegrityCheck algorithm in the definition shown in Section 3.2.
4. The comparison is not enough, revocable ABE has been studied for a long time. The authors should give the comparison with more papers, especially those published in high-quality journals/conferences.
Comments on the Quality of English LanguageThe authors should carefully proofread the paper to improve the quality of the paper. Lots of grammar mistakes and typos are in the paper. To name a few,
- algorithm Setup in Sec. 4.1, p.7: In "sk_k = { \beta_i }_{ k \in AA_k }", i is not defined before used. Maybe it should be i \in AA_k? But AA_k is an authority, not a set, and thus "k \in AA_k" does not make sense either. Same problem happens in the entire content of Sec. 4, e.g., "{s_i}_{i \in AA_k}" in Step 1 of the KeyGen algorithm.
- Step 4 of KeyGen algorithm, p.7: "AT" should be "TA", and k should be sampled from Z_n, not Z_N. Besides, "path(utag)" should be placed as a subscript.
-
Reviewer 2 Report
Comments and Suggestions for Authors
The manuscript "Revocable and traceable Undeniable Attribute-Based Encryption in Cloud-Enabled E-Health Systems" by He et al. is a commendable piece of research that significantly contributes to the field of cloud-based electronic healthcare systems. The proposed MA-RUABE scheme addresses critical issues in the current state of Ciphertext Policy Attribute-Based Encryption (CP-ABE), particularly around the facets of attribute revocation, computational capabilities, and key management. The authors have meticulously developed a framework that not only enhances computational efficiency through direct revocation features but also pioneers in introducing a multi-permission mechanism to mitigate centralized power risks in single attribute permissions.
Strengths:
-
Innovative Approach: The manuscript successfully introduces an innovative EHR sharing model based on cloud storage, which effectively identifies and prevents malicious user key leakage. The direct key revocation and multi-authority strategies are particularly noteworthy for their novel approach to enhancing security and efficiency.
-
Comprehensive Analysis: The authors provide a thorough security analysis, demonstrating the system's resilience against chosen plaintext attacks. The inclusion of experimental results to benchmark MA-RUABE against computational overhead further solidifies the manuscript's credibility.
-
Relevance and Practicality: Addressing the pressing need for robust security in electronic healthcare data, the manuscript's relevance to current technological and security challenges in healthcare IT is highly commendable.
Recommended Minor Revisions:
-
Clarity in Scheme Description: While the scheme is described in detail, certain sections might benefit from further clarification for readers not deeply familiar with the technical aspects of CP-ABE. Simplifying these explanations or providing additional context might enhance the manuscript's accessibility.
-
Comparative Analysis with Recent Works: The manuscript would be strengthened by a more detailed comparison with the most recent developments in the field. Adding a few more recent references and discussing how MA-RUABE improves or differs from these works would provide a clearer positioning of the paper in the current research landscape.
-
Discussion on Future Work: The authors might consider adding a section on potential future enhancements to MA-RUABE. Discussing prospective research directions or possible extensions of their work could provide valuable insights and indicate areas for ongoing research.
-
Formatting and Typographical Errors: There are minor formatting inconsistencies and typographical errors that should be corrected to improve the overall readability of the paper.
Conclusion: Overall, the manuscript by He et al. is a robust and well-structured piece of research that contributes significantly to the field of cloud-enabled electronic healthcare systems. The novel approach of MA-RUABE in addressing key issues in CP-ABE systems is both innovative and highly relevant. With minor revisions, particularly in terms of clarity and comparative analysis, this paper will be an excellent addition to academic discourse in this domain.
I recommend this manuscript for publication with minor revisions. The revisions suggested would enhance the paper's clarity and depth, thereby making a strong contribution to the field.
Comments on the Quality of English LanguageThe manuscript "Revocable and traceable Undeniable Attribute-Based Encryption in Cloud-Enabled E-Health Systems" by He et al. is a commendable piece of research that significantly contributes to the field of cloud-based electronic healthcare systems. The proposed MA-RUABE scheme addresses critical issues in the current state of Ciphertext Policy Attribute-Based Encryption (CP-ABE), particularly around the facets of attribute revocation, computational capabilities, and key management. The authors have meticulously developed a framework that not only enhances computational efficiency through direct revocation features but also pioneers in introducing a multi-permission mechanism to mitigate centralized power risks in single attribute permissions.
Strengths:
-
Innovative Approach: The manuscript successfully introduces an innovative EHR sharing model based on cloud storage, which effectively identifies and prevents malicious user key leakage. The direct key revocation and multi-authority strategies are particularly noteworthy for their novel approach to enhancing security and efficiency.
-
Comprehensive Analysis: The authors provide a thorough security analysis, demonstrating the system's resilience against chosen plaintext attacks. The inclusion of experimental results to benchmark MA-RUABE against computational overhead further solidifies the manuscript's credibility.
-
Relevance and Practicality: Addressing the pressing need for robust security in electronic healthcare data, the manuscript's relevance to current technological and security challenges in healthcare IT is highly commendable.
Recommended Minor Revisions:
-
Clarity in Scheme Description: While the scheme is described in detail, certain sections might benefit from further clarification for readers not deeply familiar with the technical aspects of CP-ABE. Simplifying these explanations or providing additional context might enhance the manuscript's accessibility.
-
Comparative Analysis with Recent Works: The manuscript would be strengthened by a more detailed comparison with the most recent developments in the field. Adding a few more recent references and discussing how MA-RUABE improves or differs from these works would provide a clearer positioning of the paper in the current research landscape.
-
Discussion on Future Work: The authors might consider adding a section on potential future enhancements to MA-RUABE. Discussing prospective research directions or possible extensions of their work could provide valuable insights and indicate areas for ongoing research.
-
Formatting and Typographical Errors: There are minor formatting inconsistencies and typographical errors that should be corrected to improve the overall readability of the paper.
Conclusion: Overall, the manuscript by He et al. is a robust and well-structured piece of research that contributes significantly to the field of cloud-enabled electronic healthcare systems. The novel approach of MA-RUABE in addressing key issues in CP-ABE systems is both innovative and highly relevant. With minor revisions, particularly in terms of clarity and comparative analysis, this paper will be an excellent addition to academic discourse in this domain.
I recommend this manuscript for publication with minor revisions. The revisions suggested would enhance the paper's clarity and depth, thereby making a strong contribution to the field.
Round 2
Reviewer 1 Report
Comments and Suggestions for Authors
All the comments have been addressed, except for the security model and definition issues. The security model given in Section 3.3 still not consider the collusion between the authorities. If the collusion is considered, then the security model needs to be revised such that the adversary is allowed to obtain the secret key SK_k for some AA_k. That also means that the security proof should be also modified such that the reduction algorithm give the secret key SK_k for AA_k chosen by the adversary. I wonder the security of the proposed scheme can be still proven based on the IND-CPA security of [17] if the authors revise the model into a multi-authority setting, since the scheme of [17] is design as a single authority scheme. Besides, the authors did not address the issue of Setup algorithm either. As mentioned in the review report, if there is an authority being responsible of performing Setup algorithm, then the authority knows all the master secrets, which makes the proposed scheme centralized. The problem cannot be easily solved even if we allow multiple authorities to perform Setup algorithm interactively. More precisely, if there is an authority obtain the master secret (p, q, \alpha, a, g_3) after Setup algorithm is performed, then this authority can easily compute e(g, g)^{\alpha s} = e(g, C_0), and recover the encrypted message M from C. If there is an authority who is able to decrypt all the ciphertexts in the system, then the scheme is definitely not a multi-authority system.
Comments on the Quality of English LanguageThe issues of English language have been dealt with in the revised version.
Round 3
Reviewer 1 Report
Comments and Suggestions for Authors
I accept the statement that TA should be trustworthy. However, the current architecture is not the standard multi-authority setting, in both the construction and the security model. I suggest a "future improvement" section should be added in order to summarize the issues and the shortcomings of the current version.
Comments on the Quality of English LanguageMost of the language and spelling issues have been addressed, except for the "sk_{p}ri" in the "Phase" of the dishonest AA game at p. 8. It seems that "pri" should be in the subscript of sk.
